# Tailoring Morphology and Properties of Tight Utrafiltration Membranes by Two-Dimensional Molybdenum Disulfide for Performance Improvement

**DOI:** 10.3390/membranes12111071

**Published:** 2022-10-29

**Authors:** Huali Tian, Xing Wu, Kaisong Zhang

**Affiliations:** 1Key Laboratory of Ecology of Rare and Endangered Species and Environmental Protection, College of Life Sciences, Guangxi Normal University, Ministry of Education, Guilin 541000, China; 2Key Laboratory of Urban Pollutant Conversion, Institute of Urban Environment, Chinese Academy of Sciences, Xiamen 361021, China; 3CSIRO Manufacturing, Clayton South, Victoria 3169, Australia; 4Key Laboratory of Marine Environment and Ecology, Ministry of Education, Ocean University of China, Qingdao 266100, China

**Keywords:** tight ultrafiltration membrane, molybdenum disulfide, humic acid rejection, BSA rejection

## Abstract

To enhance the permeation and separation performance of the polyethersulfone (PES) tight ultrafiltration (TUF) membrane, two-dimensional molybdenum disulfide (MoS_2_) was applied as a modifier in low concentrations. The influence of different concentrations of MoS_2_ (0, 0.25, 0.50, 1.00, and 1.50 wt%) on TUF membranes was investigated in terms of morphology, mechanical strength properties, permeation, and separation. The results indicate that the blending of MoS_2_ tailored the microstructure of the membrane and enhanced the mechanical strength property. Moreover, by embedding an appropriate amount of MoS_2_ into the membrane, the PES/MoS_2_ membranes showed improvement in permeation and without the sacrifice of the rejection of bovine serum protein (BSA) and humic acid (HA). Compared with the pristine membrane, the modified membrane embedded with 0.5 wt% MoS_2_ showed a 36.08% increase in the pure water flux, and >99.6% rejections of BSA and HA. This study reveals that two-dimensional MoS_2_ can be used as an effective additive to improve the performance and properties of TUF membranes for water treatment.

## 1. Introduction

To meet the growing demand for clean water, water recovery from wastewater by membrane technologies has been an emerging strategy [1,2]. The ultrafiltration membrane (UF) has become an interesting strategy because of its effectiveness in removing particles, microorganisms, and other organic contaminants [3]. Among them, the tight ultrafiltration (TUF) membrane (a molecular weight cut-off (MWCO) of ~1000–10000 Da [4]) has a higher retention performance on organic material. Therefore, using TUF for the removal of natural organic matter (NOM) in water has attracted more and more attention [5,6,7]. However, current TUF membranes face a critical challenge: the trade-off between water permeability and selectivity. In general, conventional polymer-based UF membranes could not break the “upper bound” between the separation factor and membrane permeability [8]. Since the steric effect is the main separation mechanism of the UF membrane [9], and the selectivity of the UF membrane is determined by the pore radius (R) and pore radius distribution (σ), while permeability is mainly determined by pore radius (R), porosity (ε) and the thickness of selectivity layer (δm). The selectivity and permeability are inversely related [10]. Therefore, conventional modification approaches make it difficult to break through the trade-off limitation.

Recently, to enhance the water flux and separation performance of the membranes, polymer matrix membranes are advanced membranes incorporating inorganic materials into an organic or a polymer matrix [11,12,13,14,15,16,17,18]. With a high surface area and adjustable interlayer spacing, two-dimensional (2D) materials such as graphene and its derivatives can create a powerful platform to accommodate convenient transport carriers and facilitate transportation [19]. For membrane modification, previous research demonstrated that graphene oxide (GO) enhances the water permeation and fouling resistance in the membrane [20,21]. Molybdenum disulfide (MoS_2_)_,_ another typical 2D material, is one of the transition-metal dichalcogenides (TMDCs) that has been studied extensively [22]. MoS_2_ occurs naturally on the earth’s crust as a molybdenum mineral, making it easier to produce on a large scale [23], and multilayered MoS_2_ can be obtained by simply exfoliating [24,25,26]. Moreover, the distance between two-layer MoS_2_ is about 0.62 nm, and the laminar channel spacing is about 0.29 nm, which is slightly larger than the size of water molecules [24,27]. In addition, MoS_2_ does not have additional functional groups, like the GO surface. Therefore, the water channels in MoS_2_ are smooth, resulting in a 2–10 times faster water passage rate compared to GO [28,29,30]. In addition, MoS_2_ exhibits good stability in a wide range of pH aqueous solutions and mechanical stability under pressure [31,32]. Therefore, MoS_2_ can be used as a suitable 2D building block for the fabrication of separation membranes with relatively fixed-size nanochannels [29]. These advantages of MoS_2_ may bring more water molecular channels to the mixed-matrix membrane and improve permeability without sacrificing membrane interception. Recently, there are only a few studies which use MoS_2_ to modify membranes. For example, in the preparation of thin film nanocomposite (TFN) film, 2D MoS_2_ was used to introduce polyimide (PA), a selective layer, to improve the salt rejection, water permeability, hydrophilicity, electronegativity, and anti-pollution properties of the membrane [24,33,34]. In addition, in the UF membrane, functionalized MoS_2_ was added to the membrane matrix to improve the membrane permeability and anti-pollution properties [35,36]. However, most studies ignored the detailed study of the regulation of membrane pores and membrane morphology by 2D MoS_2_. In addition, there are few detailed studies on the NOM removal of MoS_2_ mixed-matrix membranes [23]. In our previous study, MoS_2_ was introduced into the PES matrix, and we found that a high concentration (3.0wt%) of MoS_2_ could reduce the pore size of the membrane and improve the permeation flux of the membrane [37]. However, the high concentration of MoS_2_ was bound to burden the cost of membrane preparation. 

In this work, to reduce the adverse effect caused by the high concentration of MoS_2_ and investigate the influence of MoS_2_ on membrane performance, we fabricated a series of PES/MoS_2_ TUF membranes by the phase inversion method and used a low content of 2D MoS_2_ as additives. The morphology and filtration performance of as-prepared membranes were studied and the optimal loading of MoS_2_ in the casting solution of TUF membranes was investigated. We obtained a higher permeability membrane without reducing the rejection of humic acid (HA) and bovine serum (BSA) protein, which indicates that commercial MoS_2_ has great potential for membrane modification.

## 2. Experimental 

### 2.1. Materials

Polyethersulfone (Ultrason E6020P) was purchased from BASF (Ludwigshafen, Germany). Molybdenum disulfide (MoS_2_) (99.5% metals basis, <2 μm) and bovine serum albumin (BSA, 67 KDa) were obtained from Aladdin Chemistry Co. Ltd., Shanghai, China. N,N-Dimethylacetamide (DMAc) and polyvinylpyrrolidone (PVP) were purchased from Sinopharm Chemical Reagent Co., Ltd., Shanghai, China. Polyethylene glycol *PEG* (2000 Da, 4000 Da, 6000 Da, 8000 Da, 10,000 Da) and HA was acquired from Sigma Aldrich (Sigma–Aldrich, Inc., St. Louis, MO, USA).

### 2.2. Membrane Preparation

Different amounts of dry MoS_2_ were added to DMAc solution and ultrasonicated for 4 h for well dispersion (QSONICA SONICATORS, Newtown, CT, USA, 500W, 75%). Then, polymer (PES) and membrane pore-forming agents (PVP) were dissolved in the solvent, and the homogeneous casting solution was obtained by heating and stirring. The viscosity of the casting solution containing different concentrations of MoS_2_ additives is shown in Table 1. To prepare the membrane, the solution was cast on non-woven fabrics, and then immersed into the deionized water (DI) water bath. Finally, the prepared membrane was immersed in another water bath for 24 h to remove residues and then stored in DI water prior to use. The synthesis and filtration process diagram of PES/MoS_2_ membrane is shown in Figure 1.

### 2.3. Characterization of MoS_2_ and Membranes

The morphologies of MoS_2_ were characterized by file emission scanning electron microscopy (FESEM, HITACHIS-4800, Hitachi, Tokyo, Japan) and transmission electron microscopy (TEM, H-7650, Hitachi, Japan). The zeta potential (Zeta PALS, Malvern Instruments Ltd., Malvern, UK) was used to characterize the electronegativity of MoS_2_.

To verify that the 2D MoS_2_ sheets were incorporated into the UF membranes successfully, X-ray diffraction (XRD X’Pert, Pro, PANalytical, Almelo, Netherlands) was used to investigate the MoS_2_ and the prepared membranes in a range of 10–70°. Morphological structures of the membranes were examined by using a FESEM (HITACHIS-4800, Hitachi, Japan) and TEM (H-7650, Hitachi, Japan). The hydrophilicity of the membranes was determined using a contact angle goniometer (Dataphysics OCA20, Dataphysics, Filderstadt, Germany). A water drop with a volume of 5 μL was dropped onto the top surface of each membrane. At least three contact angles at different locations on each sample were recorded to obtain a reliable value. The mechanical properties of prepared membranes were analyzed by measuring tensile stress using tensile testing equipment (LRK-500N, NTS, Tokyo, Japan). The membrane samples were stretched at an elongation rate of 100 mmmin^−1^. The membrane was initially fixed by grips at a distance of 55 mm, after which the movable crosshead containing the load cell of 500N pulled the membrane until the membrane broke.

### 2.4. Molecular Weight Cut-off, Pore Size, Porosity, and Filtration Performance of Membranes

The MWCO of membranes was defined by polyethylene glycol (*PEG*) solutions with a concentration of 1.0 gL^−1^, and the retention rate was 90% [38]. Rejection measurements were performed using the same dead-end filtration cell (Model 8010, Millipore Corp, Burlington, MA, USA) at a pressure of 0.1 MPa after a 30 min pre-compaction at 0.15 MPa. The concentrations of *PEG* in the feed solution and permeate were measured by a total organic carbon analyzer (TOC,TOC-LCSH, Shimadzu, Tokyo, Japan) [39]. The *PEG* rejection of as-prepared membranes was calculated by Equation (1):*R* (%) = (1 − *C_p_*/*C_f_*) × 100(1)
where *C_p_* and *C_f_* were the *PEG* concentrations of permeate and feed solutions, respectively (gL^−1^).

The pore diameter of the membrane was equal to the Stokes radius (*d_s_*) of *PEG* at a 50% rejection, which could be calculated by Equation (2) [40]:(2)$$d_{s}=2\times 16.73\times 10^{-12}\times M_{PEG}^{0.557}$$

The porosity ε (%) was determined by a gravimetric method, as defined by Equation (3) [41]:(3)ε (%)=(ww−wd)DW(ww−wd)DW +WdDP×100% 
where ε is the porosity of membranes (%); *W_w_* and *W_d_* are related to the wet weight and the dry weight of the membrane (g), respectively; *D_w_* (0.998 g cm^−3^) is the density of the water; and *D_p_* (0.37 g cm^−3^) is the density of polymer.

The water flux was tested by a dead-end filtration cell with a volume capacity of 10 mL and the effective area of the membrane was 4.1 cm^2^ at room temperature (25 ± 1 °C). The samples were prepressed at 0.15 MPa for 30 min with DI water as the feed solution, ensuring that the flux reaches a steady state, and then the pure water flux of each membrane sample was recorded at 0.1 MPa by monitoring the weight change in permeate with an electronic balance (Sartorius BS224S, Sartorius AG, Goettingen, Germany). The permeate flux (*J*_0_) was calculated by Equation (4):(4)$$J_{0}=\frac{\Delta V}{A_{m}\Delta t}$$
where *J*_0_ (Lm^−2^h^−1^) was the membrane flux, Δ*V* (L) was the volume of permeated water during the period of permeation time Δ*t* (h), and *A* (m^2^) was the testing membrane area.

The filtration performances of the membranes were investigated by dead-end filtration cell and using BSA and HA as model solutes. The BSA (1.0 gL^−1^) solution in PBS (pH = 7.4), and HA (20 mgL^−1^) in DI water. The concentration of these two chemical substances was measured by ultraviolet–visible (UV–vis) spectrophotometer (Spectra Max M2, Molecular) at 280 nm (BSA) and 254 nm (HA), respectively. The permeation flux and BSA/HA rejections were calculated using Equations (1) and (4), respectively.

## 3. Results and Discussion

### 3.1. Characterization of MoS_2_

The morphology of original MoS_2_ and MoS_2_ after 4 h of sonication was observed by SEM and TEM images. As shown in Figure 2, it can be observed that the MoS_2_ flakes with ultrasonic treatment (Figure 2b) have enhanced dispersion compared to the untreated MoS_2_ flakes (Figure 2a). The original MoS_2_ comprised a multilayered structure, as shown in Figure 2c. Compared to that, the lamellar number of MoS_2_ was significantly reduced (Figure 2d). The reduced layer number was further confirmed by TEM images in Figure 2e,f.

The crystallite structure of MoS_2_ was measured by XRD analysis, and these patterns are represented in Figure 3a. The peaks at 2θ = 14.4°, 29°, 39.6°, 44.2°, 49.8°, and 60.2° were attributed to (002), (004), (103), (104), (105), and (110) planes of MoS_2_, respectively [42,43]. The zeta potential of MoS_2_ is shown in Figure 3b. The zeta potential of MoS_2_ was −31.5 ± 5.3 mV at pH 3 and gradually reduced to −41.35 ± 2.90 mV at pH 10. The decrease zeta potential of MoS_2_ was because of the oxidation of Mo. Mo can exist in the form of HMoO_4_^−^ or MoO_4_^2−^ in the aqueous solution, making MoS_2_ negatively charged [44,45].

### 3.2. Characterization of MoS_2_/PES Membrane °

The microstructure of the fabricated membranes was observed by SEM (Figure 4). Compared to the smooth surface of the pure PES membrane M0, there was some MoS_2_ exposed on the surface of PES/MoS_2_ membranes. With an increase in the concentration of MoS_2_ in casting solutions, more MoS_2_ was observed on the membrane surface. This result was associated with the influence of MoS_2_ on the membrane fabrication process. In the phase inversion process, the MoS_2_ migrated from the PES matrix to the water bath, which reduced the interfacial energy between the casting solution and the water bath. As a result, the content of MoS_2_ locally increased on the skin layer of PES membranes and more MoS_2_ can be observed on the surface of PES membranes [46]. It was observed that the M0 membrane was full of sponge-like structures. The introduction of MoS_2_ led to more finger-like pore structures in the membrane, as confirmed by the comparison between SEM images of the M0 membrane and the other PES/MoS_2_ membranes in Figure 4. Moreover, with an increase in the MoS_2_ concentration from 0 (M0) to 0.50 (M2) wt%, the finger-like structures became longer and wider. Additionally, the microvoids were closer to the skin layer. The major reason for this result was due to the addition of MoS_2_ which accelerated the instantaneous exchange of solvent and non-solvent in the casting solution, leading to the formation of microvoids in the membranes [47,48]. In addition, MoS_2_ sheets migrated to the skin layer of the membrane when the solvent and non-solvent exchange happened, which resulted in the formation of finger-like structures in the membrane interior.

The cross-sectional structures were further investigated by TEM, as shown in Figure 5. This indicates that the MoS_2_ dispersed evenly in the PES matrix in the PES/MoS_2_ membrane. With an increase in the MoS_2_ concentration, the MoS_2_ brought various porous structures to the PES matrix. This was due to the formation of cavities between 2D sheets and polymers. Interestingly, some cavities appeared in the dense skin layer of the PES/MoS_2_ membranes. By using Image J software, the thickness of the dense skin layer in each membrane was measured. This indicates that the skin layer thickness increased with the increased concentration of MoS_2_ in membranes, as M0 (3.46 μm) < M2 (3.83 μm) < M4 (4.80 μm). This may be related to the increased viscosity of the casting solution by adding MoS_2_, as shown in Table 1.

Figure 6 shows the XRD results of the PES membrane and the PES/MoS_2_ membranes. All the membranes showed typical peaks representing the amorphous region of PES in 2*θ* at 17.6°, 22.6°, and 25.9° [49]. Compared to the XRD result of the M0 membrane, there were new peaks in the spectra of PES/MoS_2_ membranes. Peaks at 2*θ*=14.4°, 29°, 39.6°, 44°, 49.8°, and 60.1° were assigned to (002), (004), (103), (104), (105) and (110) planes of MoS_2_, which were matched well with JCPDS card no. 37–1492 [50,51,52]. Moreover, the M4 membrane showed more obvious typical MoS_2_ peaks due to the high concentration of MoS_2_ in the membrane. These results indicate that MoS_2_ was successfully blended in PES/MoS_2_ membranes.

To investigate the effect of the MoS_2_ concentration on the hydrophilicity of membrane surface, the water contact angles (CAs) of the membranes were measured. It was found that the contact angle of the M0 membrane was approximately 47.9° (Figure 7a). With the concentration of the MoS_2_ increasing, the CA values of the TUF membranes increased gradually, which can be ascribed to the hydrophobic nature of the MoS_2_ [53].

The mechanical strength properties of the PES and PES/MoS_2_ membranes were expressed in terms of tensile strength and elongation at break. The result in Figure 7 indicates that the addition of MoS_2_ improved the tensile strength of the modified membranes which was in line with other studies on MoS_2_ [25]. For example, the M0 membrane was found to have a tensile strength of 36.09 MPa at elongation at a break of 7.98%. Compared to that, the M4 membrane had a tensile strength of 48.59 MPa at a break of 10.97%. In the mixed-matrix membranes, MoS_2_ could cross-link with the polymeric chains and increase the rigidity of polymeric chains. Due to this, the mechanical strength of mixed-matrix membranes was enhanced. However, the tensile strength did not show a gradual increase with increasing MoS_2_ content for membranes M1-M4. When the concentration of MoS_2_ was 0.50 wt%, the membrane M2 showed lower tensile strength than that of the membrane M1, which was due to the longer and wider finger-like structures in the membrane M2. The porous structures in the M2 membranes reduced the tensile strength.

The MWCOs of as-prepared membranes were measured, and the results are shown in Figure 8. The fabricated membranes exhibited an MWCO range between 7.80 and 9.78 kDa, which confirmed that these membranes were TUF membranes. Compared to the pristine PES membrane with 7.83 kDa of MWCO, the MWCO increased to 9.78 kDa for the membrane M1 and 9.67 kDa for the membrane M2, but then dropped to 7.83 kDa for the membrane M3 and 7.80 kDa for the membrane M4. The corresponding pore radius of the TUF membranes is shown in Table 2. The results indicate that the pore sizes of the membranes increased as the concentration of MoS_2_ in the casting solutions was 0 to 0.50 wt%, while higher contents of nanofillers (1.00–1.50 wt%) may lead to the MoS_2_ having a strong interaction with PES and DMAc, which reflected an increase in the viscosity of the casting solution. These effects led to a delay in PES precipitation, a decrease in the diffusion rate of water in the PES matrix, and a delay in liquid–liquid stratification, which inhibited the synthesis of macropores [54,55] and consequently reduced the membrane pore size.

Table 2 also shows the pure water flux of the prepared TUF membranes. The result indicates that the pure water flux of membranes initially improved with the increased content of MoS_2_ when the concentration was 0 to 0.50 wt%. The pure water flux of the M0 membrane without MoS_2_ was found to be 72.37 Lm^−2^h^−1^, and it increased to 91.86 Lm^−2^h^−1^ for the M1 membrane. The water flux permeability of UF membranes was determined by properties including the porosity, pore size, and thickness of the membranes’ skin layer. Compared to the M0 membrane, the M1 membrane has similar porosity but a larger MWCO and mean pore radius. Therefore, the improved permeability in the M1 membrane was potentially dominated by the influence of the large pore size. Similar results could also be found when comparing the performance of the M2 and M3 membranes. Compared to the M0 membrane, the M2 membrane exhibited both improved porosity and pore size, which resulted in 1.36 times higher water flux (98.48 Lm^−2^h^−1^) than that of the M0 membrane. However, a further increase in the MoS_2_ concentration increased the membrane surface hydrophobicity, increased the thickness of the skin layer, and the reduced the membrane pore size, which consequently decreased the water flux of the M4 membrane to 80.18 Lm^−2^h^−1^.

Figure 9 shows the filtration performance of as-prepared TUF membranes by using BSA solution as the feed solution. The result indicates that all as-prepared PES/MoS_2_ membranes have higher BSA rejections than the pristine membrane (M0), and all of them were over 99.50%. This was related to the fact that the negative charge brought by MoS_2_ increased the negative charge on the membrane surface. The permeation flux of the BSA solution for the M0 membrane was 62.37 Lm^−2^h^−1^. The water flux increased to 68.29 Lm^−2^h^−1^ for the M1 membrane, and the rejection increased to 99.85%. As a function of the MoS_2_ concentration, the permeation flux decreased gradually. An initial increase in the permeation flux was attributed to the pore size change, while the following decrease was due to the increased hydrophobicity of the membranes [56].

The filtration performance of prepared TUF membranes was further investigated by using HA aqueous solution as the feed solution. This result shows in Figure 10, indicates that all the membranes have excellent separation performance for HA with the rejection rate between 99.21 and 99.60%. Moreover, the permeation flux of the HA solution for the M1 membrane (0.25 wt%) was 77.24 Lm^−2^h^−1^, which was 33.4% higher than that of the M0 membrane (57.90 Lm^−2^h^−1^). This increase was primarily dependent on the change in pore size. However, it was also found that the effective pore sizes of the M1 and M3 membranes were almost the same and the permeation was changed. This phenomenon may be caused by an increase in the surface hydrophobicity of the membrane due to an increase in the content of MoS_2_. HA adsorption onto a PES membrane has been confirmed in other research and mainly correlated to the hydrophobic groups of molecules [57]. The effect of adsorption reduced the permeability of the membrane. The results show that further research on enhanced hydrophilicity of 2D MoS_2_ would improve the antifouling performance of the mixed-matrix membrane. Table 3 compares the HA separation performance of the PES/MoS_2_ membrane with some other membranes in the previous literature [23,58,59,60,61,62], and indicates that the PES/MoS_2_ membrane has better HA selectivity compared to UF membranes and improved water permeability compared to NF membranes with similar selectivity properties.

## 4. Conclusions

In this work, 2D MoS_2_ was blended into the PES matrix to enhance the performance and properties of membranes. A series of PES/MoS_2_ membranes were fabricated and their morphologies and separation performance were investigated. The results indicate that the mechanical strength property of membranes was improved by the addition of MoS_2_. SEM images proved the presence of MoS_2_ in the mixed-matrix membrane, and the finger-like macrovoid appeared. Further, the dispersed morphology was given by TEM. These changes enhanced the permeability of both pure water and the BSA/HA solution. By optimizing the concentration of MoS_2_, the PES/MoS_2_ membrane with 0.50 wt% MoS_2_ exhibited a 36.08% increase in the water flux compared to the pristine TUF membrane, and excellent rejection properties to BSA (99.85%) and HA (99.60%). This study exhibited the feasibility of applying low concentrations of MoS_2_ to tune the structure and properties of TUF membranes, which could be an effective strategy to fabricate high-performance TUF membranes.

## Figures and Tables

**Figure 1 membranes-12-01071-f001:**
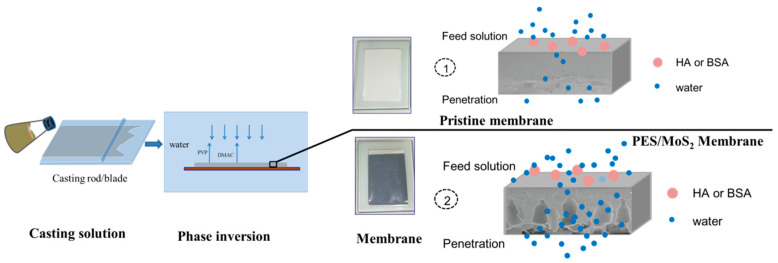
Synthesis and filtration process diagram of PES/MoS_2_ membrane.

**Figure 2 membranes-12-01071-f002:**
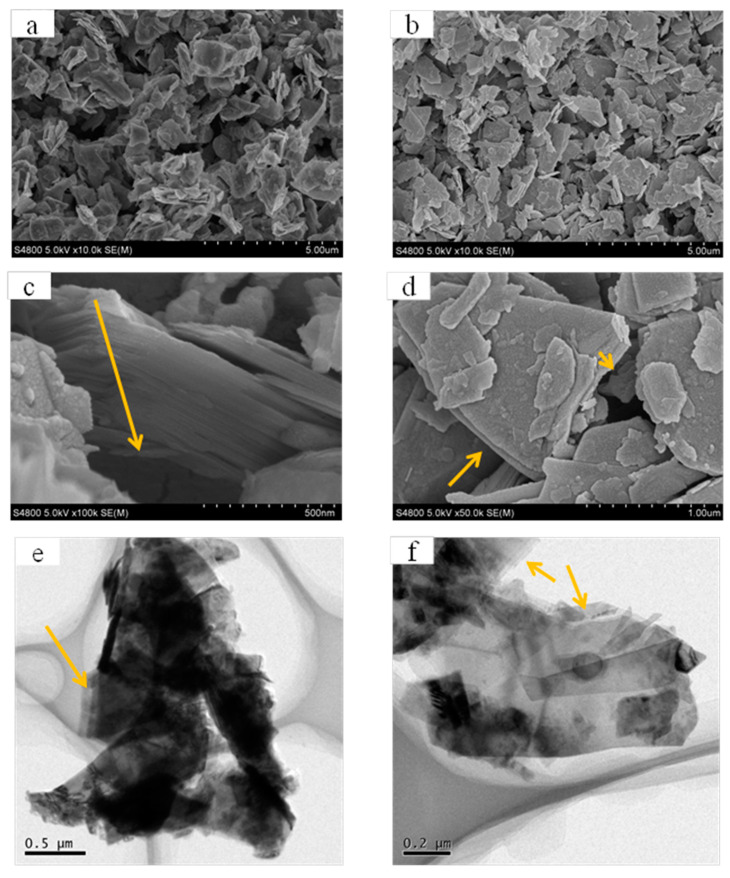
SEM images of original commercial MoS_2_ (**a**,**c**) and MoS_2_ after 4 h of sonication (**b**,**d**), TEM images of original commercial MoS_2_ (**e**), and MoS_2_ after 4 h of sonication (**f**).

**Figure 3 membranes-12-01071-f003:**
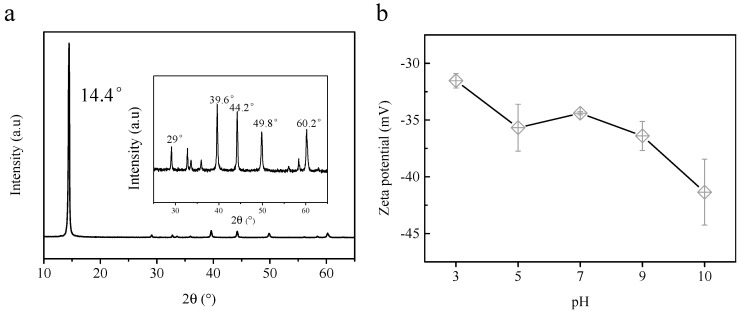
X-ray diffraction (XRD) characteristic (**a**) and zeta potential (**b**) of MoS_2_.

**Figure 4 membranes-12-01071-f004:**
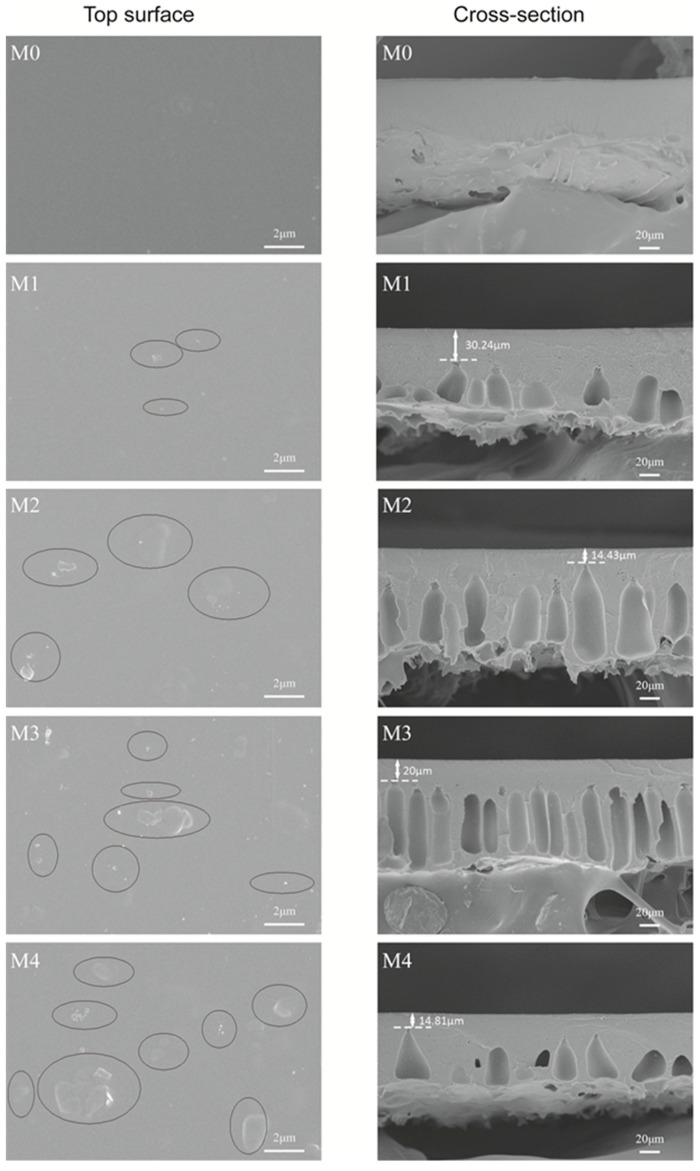
SEM surface and cross-sectional images of PES and PES/MoS_2_ membranes.

**Figure 5 membranes-12-01071-f005:**
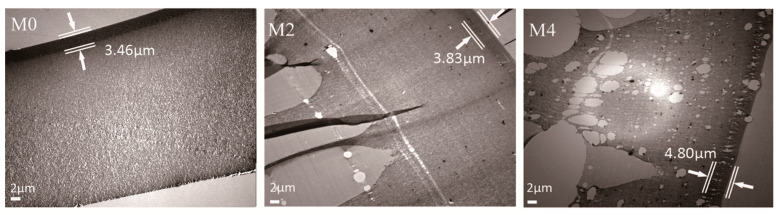
Cross-sectional TEM images of the M0, M2 and M4 membranes.

**Figure 6 membranes-12-01071-f006:**
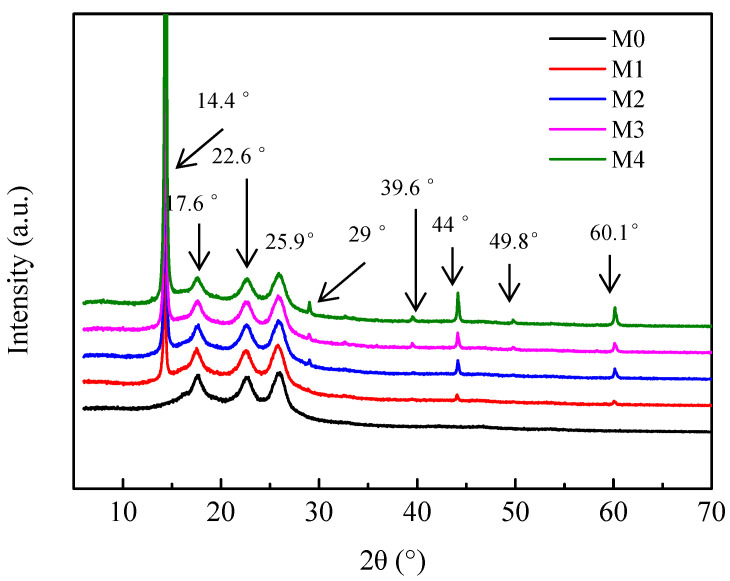
XRD patterns of PES membrane and PES/MoS_2_ membranes.

**Figure 7 membranes-12-01071-f007:**
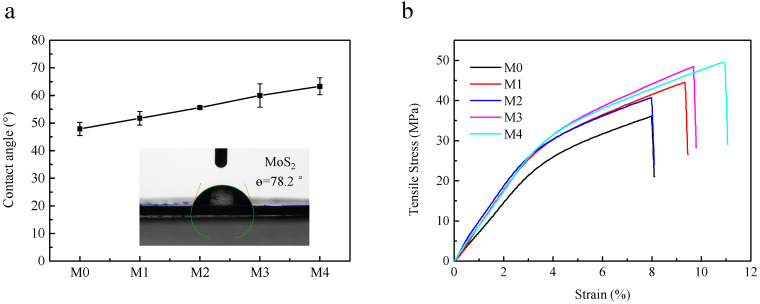
Water contact angle values (**a**) and the stress–strain behaviors (**b**) of PES and PES/MoS_2_ membranes.

**Figure 8 membranes-12-01071-f008:**
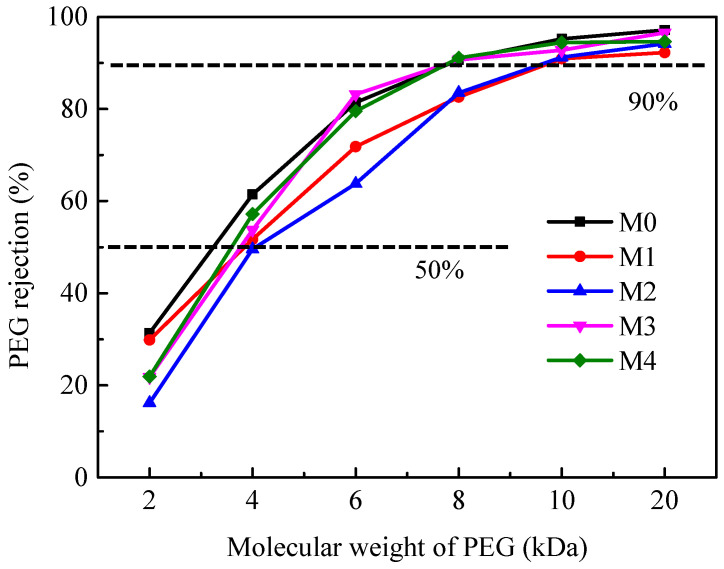
Molecular weight cut-off of PES/MoS_2_ membranes.

**Figure 9 membranes-12-01071-f009:**
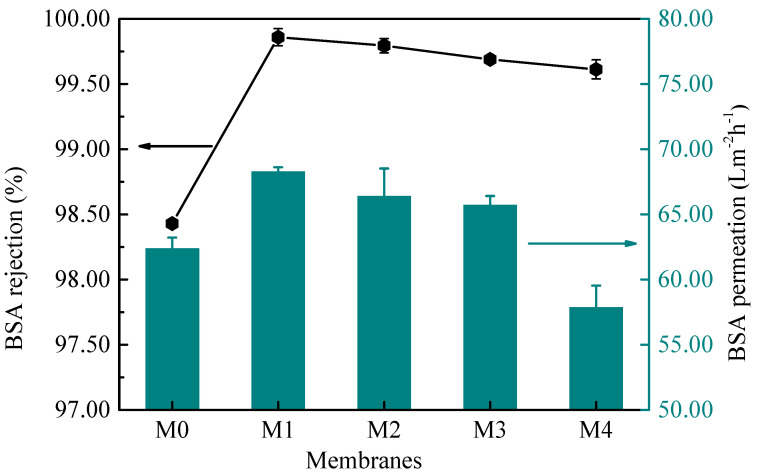
BSA rejection of TUF membranes with different concentrations of MoS_2_.

**Figure 10 membranes-12-01071-f010:**
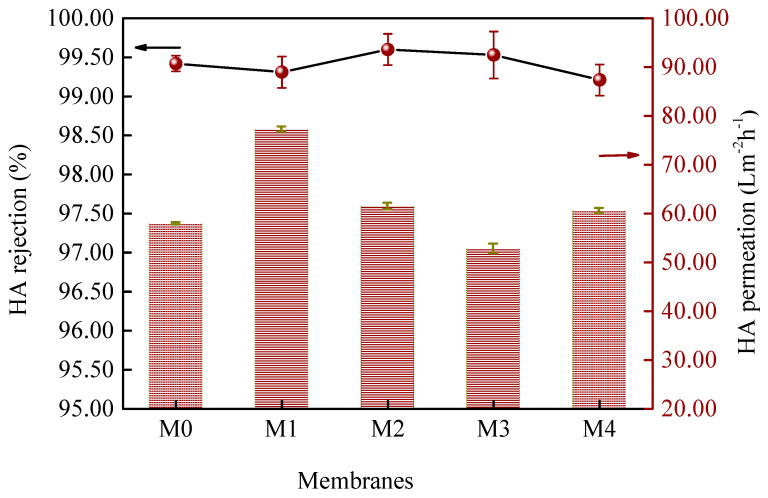
HA rejection of prepared membranes with different concentrations of MoS_2_.

**Table 1 membranes-12-01071-t001:** The composition of PES and PES/MoS_2_ casting solutions.

Membrane	DMAc (wt%)	MoS_2_ (wt%)	Viscosity (Pa.s)
M0	61.00	0.00	53.5 ± 0.1
M1	60.75	0.25	64.1 ± 1.0
M2	60.50	0.50	81.5 ± 0.6
M3	60.00	1.00	90.4 ± 0.6
M4	59.50	1.50	115.5 ± 2.6

**Table 2 membranes-12-01071-t002:** Effect of different concentrations of 2D MoS_2_ on membrane porosity, mean pore radius, and water flux.

Membrane	Porosity(%)	Mean Pore Radius (nm)	Pure Water Flux (Lm^−2^h^−1^)
M0	52.44 ± 0.15	1.51	72.37 ± 0.14
M1	51.98 ± 0.86	1.66	91.86 ± 5.79
M2	53.38 ± 0.31	1.71	98.48 ± 3.82
M3	53.50 ± 0.62	1.64	91.00 ± 6.75
M4	52.23 ± 0.98	1.60	80.18 ± 6.01

**Table 3 membranes-12-01071-t003:** Comparison of the HA separation performance of the PES/MoS_2_ membrane with some other membranes.

Membrane	Rejection (%) UV254	Permeation Flux (Lm^−2^h^−1^ bar^−1^)	Pure Water Flux (Lm^−2^h^−1^ bar^−1^)	Membrane Type	Ref
PSf/GO-Fe_3_O_4_	84 ± 2	156.99	-	UF	[58]
PSf	89 ± 2	51.78	-	UF	[58]
PVDF/PFSA-g-GO	79.6		587.4	UF	[59]
PES/GO	85.3–93.9	-	~36–108	UF	[60]
PPA-BN-4	97.91	-	14.24	NF	[61]
PES-PPA-OH-MoS_2_	99.20	-	14.023	NF	[23]
NF270	99.4	-	16	NF	[62]
PES/MoS_2_ (M2 membrane)	99.60	61.24	98.48	UF	This work

## Data Availability

The data presented in this study are available on request from the corresponding author.

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
