# Peer review of "Tailoring Morphology and Properties of Tight Utrafiltration Membranes by Two-Dimensional Molybdenum Disulfide for Performance Improvement"

_membranes, 2022, doi:10.3390/membranes12111071_

Round 1

Reviewer 1 Report

Review comments:

This work reports tight ultrafiltration membranes incorporated with molybdenum disulfide nanosheets, the obtained UF membranes were systematically characterized and tested for the removal of BSA and HA. The manuscript was well organized and written, I consider that this work could be publishable if the authors can deal with the following issues.

1. The authors should clearly point out the novelty of this work in the introduction section.

2. It is recommended that the authors should add some explanation why the conventional modification approaches cannot break the trade-off limitation, what are the challenges? Why select MoS2 as the additive instead of other more hydrophilic nanomaterials?

3. Does the sonication methods are normally or effectively used to exfoliate bulk MoS2 to obtain lamellar nanosheets? It seems that the MoS2 used in this work was still in a form of a multilayer, thus it may be hard to say it was nanosheets.

4. What are the methods to determine membrane porosity? It should be added to the methods section.

5. The authors considered that the mean pore size of M1 (i.e., 1.66 nm) larger than that of the pristine membrane M0 (i.e., 1.51 nm) was the primary reason. However, the authors should consider the porosity of those two membranes (M1<M0). Additionally, does the pore size between M1 and M0 have significant statistical differences? If not, the reasons why the M1 membrane with an increased water permeability should be carefully explained again.

6. The literature data in Table 3 should be carefully selected due to this work was related to the UF membrane. It is suggested to replace the relevant NF membrane data.

Author Response

Response to reviewers

Reviewer #1:

This work reports tight ultrafiltration membranes incorporated with molybdenum disulfide nanosheets, the obtained UF membranes were systematically characterized and tested for the removal of BSA and HA. The manuscript was well organized and written, I consider that this work could be publishable if the authors can deal with the following issues.

  1. The authors should clearly point out the novelty of this work in the introduction section.

Response:

We appreciate the positive comment from the reviewer and the suggestion to further improve the quality of our work. The paragraph in Introduction has been revised to highlight the novelty of this study, as:

“Molybdenum disulfide (MoS2), another typical 2D material, is one of the transition-metal dichalcogenides (TMDCs) that have been studied extensively [1]. MoS2 occurs naturally on the earth's crust as a molybdenum mineral, making it easier to produce on a large scale [2], and few layers MoS2 can be obtained by simply exfoliating [3]. Moreover, the distance between two-layer MoS2 is about 0.62 nm, and the laminar channel spacing is about 0.29 nm, which is slightly larger than the size of water molecules [3, 4]. In addition, MoS2 does not have additional functional groups like GO surface. Therefore, the water channels in MoS2 are smooth, resulting in 2-10 times faster water passage rate compared to GO [5-7]. Besides, the MoS2 exhibits good stability in a wide range of pH aqueous solutions and mechanical stability under pressure [8, 9]. Therefore, MoS2 would be used as a suitable 2D building block for the fabrication of separation membranes with relatively fixed-size nanochannels [6]. These advantages of MoS2 may bring more water molecular channels to the mixed matrix membrane and improve permeability without sacrificing membrane interception. Recently, there are a few studies that applied MoS2 to modify membranes. For example, in the preparation of thin film nanocomposite (TFN) film, 2D MoS2 was used to introduce into polyimide (PA) selective layer to improve the salt rejection, water permeability, hydrophilicity, electronegativity and anti-pollution properties of the membrane [3, 10, 11]. And in UF membrane, functionalized MoS2 was added to the membrane matrix to improve the membrane permeability and anti-pollution properties [12, 13]. However, most studies ignored the detailed study of the regulation of membrane pores and membrane morphology by 2D MoS2. In addition, there are few detailed studies on NOM removal of MoS2 mixed matrix membrane[2]. In our previous study, MoS2 was introduced into PES matrix, and we found that a high concentration (3.0wt%) of MoS2 could reduce the pore size of the membrane and improve the permeation flux of the membrane [14]. However, the high concentration of MoS2 was bound to burden the cost of membrane preparation.

In this work, to reduce the adverse effect caused by a high concentration of MoS2 and investigate the influence of MoS2 on membrane performance, we fabricated a series of PES/MoS2 TUF membranes by phase inversion method and using a low content of 2D MoS2 as additives. The morphology and filtration performance of as-prepared membranes were studied and the optimal loading of MoS2 in the casting solution of TUF membranes was investigated. We obtain a higher permeability membrane without reducing the rejection of HA and bovine serum (BSA) protein, which indicated that commercial MoS2 has great potential for membrane modification.”

  1. It is recommended that the authors should add some explanation why the conventional modification approaches cannot break the trade-off limitation, what are the challenges? Why select MoS2 as the additive instead of other more hydrophilic nanomaterials?

Response:

We thank the reviewer’s comment. We have supplemented the explanation why the conventional modification approaches cannot break the trade-off limitation in the revised manuscript. It was shown as:

“In general, conventional polymer-based UF membranes could not break the “upper bound” between separation factor and membrane permeability [15]. Since the steric effect is the main separation mechanism of UF membrane [16], and the selectivity of UF membrane is determined by pore radius (R) and pore radius distribution (σ), while permeability is mainly determined by pore radius (R), porosity (ε) and thickness of selectivity layer (δm). The selectivity and permeability are inversely related [17]. Therefore, the conventional modification approaches is difficult to break through the trade-off limitation.”

The discussion about “Why select MoS2 as the additive instead of other more hydrophilic nanomaterials?” has been revised in the introduction:

“Molybdenum disulfide (MoS2), another typical 2D material, is one of the transition-metal dichalcogenides (TMDCs) that have been studied extensively [1]. MoS2 occurs naturally on the earth's crust as a molybdenum mineral, making it easier to produce on a large scale [2], and few layers MoS2 can be obtained by simply exfoliating [3]. Moreover, the distance between two-layer MoS2 is about 0.62 nm, and the laminar channel spacing is about 0.29 nm, which is slightly larger than the size of water molecules [3, 4]. In addition, MoS2 does not have additional functional groups like GO surface. Therefore, the water channels in MoS2 are smooth, resulting in 2-10 times faster water passage rate compared to GO [5-7]. Besides, the MoS2 exhibits good stability in a wide range of pH aqueous solutions and mechanical stability under pressure [8, 9]. Therefore, MoS2 would be used as a suitable 2D building block for the fabrication of separation membranes with relatively fixed-size nanochannels [6]. These advantages of MoS2 may bring more water molecular channels to the mixed matrix membrane and improve permeability without sacrificing membrane interception. ”

  1. Does the sonication methods are normally or effectively used to exfoliate bulk MoS2 to obtain lamellar nanosheets? It seems that the MoS2 used in this work was still in a form of a multilayer, thus it may be hard to say it was nanosheets.

Response:

We appreciate the reviewer’s comment. MoS2 sheets can be obtained from the bulk crystal via a variety of ways [18, 19]. A number of researches have proven the feasibility to exfoliate MoS2 nanosheets from bulk materials, especially the liquid-exfoliated method assisted with sonication in the preparation of polymer-based nanocomposites [20, 21]. The references have been added into the revised manuscript. We also thank the reviewer to point out the inaccurate word “nanosheet”. In the revised manuscript, we deleted the word “nanosheet”.

  1. What are the methods to determine membrane porosity? It should be added to the methods section.

Response:

We thank the reviewer for this comment. The methods to determine membrane porosity have been added to the methods section as:

“The porosity ε (%) was determined by a gravimetric method, as defined in Equation 3[22]:

                 (3)

where ε is the porosity of membranes (%), Ww and Wd represent to the wet weight and the dry weight of the membrane (g) respectively, Dw (0.998 g cm-3) is the density of the water and Dp (0.37 g cm-3) is the density of polymer.

  1. The authors considered that the mean pore size of M1 (i.e., 1.66 nm) larger than that of the pristine membrane M0 (i.e., 1.51 nm) was the primary reason. However, the authors should consider the porosity of those two membranes (M1<M0). Additionally, does the pore size between M1 and M0 have significant statistical differences? If not, the reasons why the M1 membrane with an increased water permeability should be carefully explained again.

Response:

We appreciate the reviewer’s comment and totally agree that both porosity and pore size are critical to determine the water permeability of membranes. In our work, the porosity of the M1 membrane (52.44±0.15%) was similar to the M1 membrane (51.98±0.86%), especially when considering the error bar. However, the MWCO test indicated that the M1 membrane’s MWCO was 9.78 kDa, which was higher than the 7.83 kDa of the M0 membrane. Moreover, although the M0 and M1 membranes had similar porosity, it could be found from the SEM image that the M1 membrane had obvious larger pores (Figure. 4). Therefore, we considered the larger pore size in the M1 membrane was the major reason for the improved permeability. If we compared the performance of the M2 vs M3 membranes and M0 vs M4 (similar porosity but different pore size), we could also observe that the membrane with larger pore size had a higher permeability. We also totally agreed with the reviewer that the influence of porosity could not be ignored. Therefore, in our revised manuscript, we also mentioned the importance of porosity:

“The water flux permeability of UF membranes is determined by properties including porosity, pore size and the thickness of skin layer of the membranes. Compared to the M0 membrane, the M1 membrane has similar porosity but larger MWCO and mean pore radius. Therefore, the improved permeability in the M1 membrane was potentially dominated by the influence of the large pore size. Similar results could also be found when comparing the performance of the M2 and M3 membranes. Compared to the M0 membrane, the M2 membrane exhibited both improved porosity and pore size, which resulted in 1.36 times higher water flux (98.48 Lm-2h-1) than that of the M0 membrane. However, further increase the MoS2 concentration resulted in increasing membrane surface hydrophobicity, the thickness of skin layer and the reduced membrane pore size, which consequently decreased the water flux of the M4 membrane to 80.18 Lm-2h-1.”

  1. The literature data in Table 3 should be carefully selected due to this work was related to the UF membrane. It is suggested to replace the relevant NF membrane data.

Response:

We thank the reviewer for this comment. In Table 3, we tried to highlight the performance of the as-prepared PES/MoS2 TUF membrane by exhibiting its better selectivity compared to other previously reported UF membranes and its high permeability compared to previous reported NF membranes with similar HA rejection properties.

The discussion has been revised as:

“Table 3 compares the HA separation performance of PES/ MoS2 membrane with some other membranes in the previous literature [2, 23-27], and indicated that the PES/ MoS2 membrane has better HA selectivity compared to UF membranes and improved water permeability compared to NF membranes with similar selectivity properties.”

If the reviewer still thinks it is not inappropriate to list the NF data, we could delete them.

Reference:

[1] R.G.a.Q. Zhang, Few-Layer MoS2 A Promising Layered Semiconductor, ACS Nano, 8 (2014) 4074-4099.

[2] D.S. Mallya, G. Yang, W. Lei, S. Muthukumaran, K. Baskaran, Functionalized MoS2 nanosheets enabled nanofiltration membrane with enhanced permeance and fouling resistance, Environmental Technology & Innovation, 27 (2022) 102719.

[3] Y. Li, S. Yang, K. Zhang, B. Van der Bruggen, Thin film nanocomposite reverse osmosis membrane modified by two dimensional laminar MoS2 with improved desalination performance and fouling-resistant characteristics, Desalination, 454 (2019) 48-58.

[4] R. Dai, H. Han, T. Wang, X. Li, Z. Wang, Enhanced removal of hydrophobic endocrine disrupting compounds from wastewater by nanofiltration membranes intercalated with hydrophilic MoS2 nanosheets: Role of surface properties and internal nanochannels, Journal of Membrane Science, 628 (2021) 119267.

[5] Z. Wang, Q. Tu, S. Zheng, J.J. Urban, S. Li, B. Mi, Understanding the Aqueous Stability and Filtration Capability of MoS2 Membranes, Nano Letters, 17 (2017) 7289-7298.

[6] Z. Wang, B. Mi, Environmental Applications of 2D Molybdenum Disulfide (MoS2) Nanosheets, Environmental science & technology, 51 (2017) 8229-8244.

[7] B.Y. Guo, S.D. Jiang, M.J. Tang, K. Li, S. Sun, P.Y. Chen, S. Zhang, MoS2 Membranes for Organic Solvent Nanofiltration: Stability and Structural Control, The journal of physical chemistry letters, 10 (2019) 4609-4617.

[8] M. Deng, K. Kwac, M. Li, Y. Jung, H.G. Park, Stability, Molecular Sieving, and Ion Diffusion Selectivity of a Lamellar Membrane from Two-Dimensional Molybdenum Disulfide, Nano Lett, 17 (2017) 2342-2348.

[9] J.W. Jiang, Z. Qi, H.S. Park, T. Rabczuk, Elastic bending modulus of single-layer molybdenum disulfide (MoS2): finite thickness effect, Nanotechnology, 24 (2013) 435705.

[10] M.-Q. Ma, C. Zhang, C.-Y. Zhu, S. Huang, J. Yang, Z.-K. Xu, Nanocomposite membranes embedded with functionalized MoS2 nanosheets for enhanced interfacial compatibility and nanofiltration performance, Journal of Membrane Science, 591 (2019) 117316.

[11] H. Zhang, X.-Y. Gong, W.-X. Li, X.-H. Ma, C.Y. Tang, Z.-L. Xu, Thin-film nanocomposite membranes containing tannic acid-Fe3+ modified MoS2 nanosheets with enhanced nanofiltration performance, Journal of Membrane Science, 616 (2020) 118605.

[12] X. Liang, P. Wang, J. Wang, Y. Zhang, W. Wu, J. Liu, B. Van der Bruggen, Zwitterionic functionalized MoS2 nanosheets for a novel composite membrane with effective salt/dye separation performance, Journal of Membrane Science, 573 (2019) 270-279.

[13] M.S. Sri Abirami Saraswathi, D. Rana, P. Vijayakumar, S. Alwarappan, A. Nagendran, Tailored PVDF nanocomposite membranes using exfoliated MoS2 nanosheets for improved permeation and antifouling performance, New Journal of Chemistry, 41 (2017) 14315-14324.

[14] Mixed matrix polyethersulfone tight ultrafiltration (TUF) membrane with improved dye removal by physical blending of 2D MoS2, Desalination and Water Treatment, H.Tian,X.Wu,K.Zhang (2020) 1-14.

[15] A. Mehta, A.L. Zydney, Permeability and selectivity analysis for ultrafiltration membranes, Journal of Membrane Science, 249 (2005) 245-249.

[16] C.-E. Lin, J. Wang, M.-Y. Zhou, B.-K. Zhu, L.-P. Zhu, C.-J. Gao, Poly(m-phenylene isophthalamide) (PMIA): A potential polymer for breaking through the selectivity-permeability trade-off for ultrafiltration membranes, Journal of Membrane Science, 518 (2016) 72-78.

[17] A.L.Z. S. Mochizuki, Theoretical analysis of pore size distribution effects on membrane transport, J. Membr. Sci, 82 (1993) 211-227.

[18] J.N.L. Coleman, M.; O’Neill, A.; Bergin, S. D.; King, P. J.;Khan, U.; Young, K.;Gaucher, A.; De, S.; Smith, R. J.; Shvets, I. V.;Arora, S. K.; Stanton, G.; Kim, H. Y.;Lee, K.; Kim, G. T.; Duesberg, G.S.; Hallam, T.; Boland, J. J.; Wang, J. J.; Donegan, J. F.; Grunlan, J. C.;Moriarty, G.; Shmeliov, A.; Nicholls, R. J.; Perkins, J. M.; Grieveson, E.M.; Theuwissen, K.; McComb, D. W.; Nellist, P. D.; Nicolosi, V. , Two-Dimensional Nanosheets Produced by Liquid Exfoliation of Layered Materials Science, (2011) 568-571.

[19] Y. Yao, L. Tolentino, Z. Yang, X. Song, W. Zhang, Y. Chen, C.-p. Wong, High-Concentration Aqueous Dispersions of MoS2, Advanced Functional Materials, 23 (2013) 3577-3583.

[20] X. Feng, X. Wang, W. Xing, K. Zhou, L. Song, Y. Hu, Liquid-exfoliated MoS2 by chitosan and enhanced mechanical and thermal properties of chitosan/MoS2 composites, Composites Science and Technology, 93 (2014) 76-82.

[21] X. Feng, W. Xing, H. Yang, B. Yuan, L. Song, Y. Hu, K.M. Liew, High-Performance Poly(ethylene oxide)/Molybdenum Disulfide Nanocomposite Films: Reinforcement of Properties Based on the Gradient Interface Effect, ACS applied materials & interfaces, 7 (2015) 13164-13173.

[22] E. Salimi, A. Ghaee, A.F. Ismail, Improving Blood Compatibility of Polyethersulfone Hollow Fiber Membranes via Blending with Sulfonated Polyether Ether Ketone, Macromolecular Materials and Engineering, 301 (2016) 1084-1095.

[23] P.V. Chai, E. Mahmoudi, Y.H. Teow, A.W. Mohammad, Preparation of novel polysulfone-Fe3O4/GO mixed-matrix membrane for humic acid rejection, Journal of Water Process Engineering, 15 (2017) 83-88.

[24] X. Liu, H. Yuan, C. Wang, S. Zhang, L. Zhang, X. Liu, F. Liu, X. Zhu, S. Rohani, C. Ching, J. Lu, A novel PVDF/PFSA-g-GO ultrafiltration membrane with enhanced permeation and antifouling performances, Separation and Purification Technology, 233 (2020) 116038.

[25] K.H. Chu, Y. Huang, M. Yu, J. Heo, J.R.V. Flora, A. Jang, M. Jang, C. Jung, C.M. Park, D.-H. Kim, Y. Yoon, Evaluation of graphene oxide-coated ultrafiltration membranes for humic acid removal at different pH and conductivity conditions, Separation and Purification Technology, 181 (2017) 139-147.

[26] S. Abdikheibari, W. Lei, L.F. Dumée, N. Milne, K. Baskaran, Thin film nanocomposite nanofiltration membranes from amine functionalized-boron nitride/polypiperazine amide with enhanced flux and fouling resistance, Journal of Materials Chemistry A, 6 (2018) 12066-12081.

[27] H. Song, J. Shao, Y. He, J. Hou, W. Chao, Natural organic matter removal and flux decline with charged ultrafiltration and nanofiltration membranes, Journal of Membrane Science, 376 (2011) 179-187.

Reviewer 2 Report

This article is comprehensive, logically organized, and contains valuable information on the tailoring morphology and properties of tight ultra-filtration (TUF) membranes by Molybdenum disulfide (MoS2) nanosheets for performance improvement. The authors did excellent research on investigating the two-dimensional MoS2 nanosheets in low concentrations that were added as modifiers to enhance the permeation and separation performance of polyethersulfone (PES) TUF membranes for water treatment applications.

To improve the manuscript, the authors should take the following considerations:

(1) The authors presented the composition of PES and PES-MoS2 casting solutions in Table 1. The authors presented the viscosity (mpas). To avoid many numbers in Table 1, it is suggested that the authors should present viscosity data in Pascal second (Pa.s) instead of the milliPascal second (mPa s). For example, 115533.33±2610.23 mpas should be written as 115.5±2.6 Pa.s

(2) The authors presented the X-ray diffraction (XRD) characteristic of MoS2 in Figure 3. What is the JCPDS card No of MoS2? It is suggested that the authors should place the XRD patterns of the JCPDS card for comparison purposes.

(3) Title: “Utrafiltraion” should be written as “Ultrafiltration”.

The submitted manuscript has significant scientific insights and the conclusions are soundly supported by the experimental data. However, the present submission requires minor revisions before being considered for publication in the Special Issue: Membrane Applications, UF/NF/RO Membranes for Wastewater Treatment and Reuse, in the esteemed Membranes in its current condition.

Author Response

Response to reviewers

Reviewer #2:

This article is comprehensive, logically organized, and contains valuable information on the tailoring morphology and properties of tight ultra-filtration (TUF) membranes by Molybdenum disulfide (MoS2) nanosheets for performance improvement. The authors did excellent research on investigating the two-dimensional MoS2 nanosheets in low concentrations that were added as modifiers to enhance the permeation and separation performance of polyethersulfone (PES) TUF membranes for water treatment applications.

To improve the manuscript, the authors should take the following considerations:

(1) The authors presented the composition of PES and PES-MoS2 casting solutions in Table 1. The authors presented the viscosity (mpas). To avoid many numbers in Table 1, it is suggested that the authors should present viscosity data in Pascal second (Pa.s) instead of the milliPascal second (mPa s). For example, 115533.33±2610.23 mpas should be written as 115.5±2.6 Pa.s

Response:

We thank the reviewer for this comment. We have presented viscosity data in Pa.s instead of the mPa s in the revised manuscript. The table1 has been revised as:

Table 1. The composition of PES and PES-MoS2 casting solutions.

Membrane

DMAc (wt%)

MoS2 (wt%)

Viscosity ( Pa.s )

M0

61.00

0.00

53.5±0.1

M1

60.75

0.25

64.1±1.0

M2

60.50

0.50

81.5±0.6

M3

60.00

1.00

90.4±0.6

M4

59.50

1.50

115.5±2.6

(2) The authors presented the X-ray diffraction (XRD) characteristic of MoS2 in Figure 3. What is the JCPDS card No of MoS2? It is suggested that the authors should place the XRD patterns of the JCPDS card for comparison purposes.

 Response:

 We thank the reviewer’s comment. We placed the XRD patterns of the JCPDS card (no. 37-1492) in the manuscript as:

“Peaks at 2θ=14.4°, 29°, 39.6°, 44°, 49.8°and 60.1° were assigned to (002), (004), (103), (104), (105) and (110) planes of MoS2 which was matched well with JCPDS card no. 37-1492 [1-3].”

(3) Title: “Utrafiltraion” should be written as “Ultrafiltration”.

Response:

We thank the reviewer pointing this mistake out. We have corrected “Utrafiltraion” as “Ultrafiltration” .

Reference:

[1] J. Zheng, H. Zhang, S. Dong, Y. Liu, C.T. Nai, H.S. Shin, H.Y. Jeong, B. Liu, K.P. Loh, High yield exfoliation of two-dimensional chalcogenides using sodium naphthalenide, Nature communications, 5 (2014) 2995.

[2] M. Saraswathi, D. Rana, A. Nagendran, S. Alwarappan, Custom-made PEI/exfoliated-MoS2 nanocomposite ultrafiltration membranes for separation of bovine serum albumin and humic acid, Mater Sci Eng C Mater Biol Appl, 83 (2018) 108-114.

[3] Z. Cheng, Y. Xiao, W. Wu, X. Zhang, Q. Fu, Y. Zhao, L. Qu, All-pH-Tolerant In-Plane Heterostructures for Efficient Hydrogen Evolution Reaction, ACS nano, (2021).
